# CaseGen: A Benchmark for Multi-Stage Legal Case Documents Generation

## Abstract

Legal case documents play a critical role in judicial proceedings. As the number of cases continues to rise, the reliance on manual drafting of legal case documents is facing increasing pressure and challenges. The development of large language models (LLMs) offers a promising solution for automating document generation. However, existing benchmarks fail to fully capture the complexities involved in drafting legal case documents in real-world scenarios. To address this gap, we introduce CaseGen, the benchmark for multi-stage legal case documents generation in the Chinese legal domain. CaseGen is based on 500 real case samples annotated by legal experts and covers seven essential case sections. It supports four key tasks: drafting defense statements, writing trial facts, composing legal reasoning, and generating judgment results. To the best of our knowledge, CaseGen is the first Chinese benchmark designed to evaluate LLMs in the context of legal case document generation. To ensure an accurate and comprehensive evaluation, we design the LLM-as-a-judge evaluation framework and validate its effectiveness through human annotations. We evaluate several widely used general-domain LLMs and legal-specific LLMs, highlighting their limitations in case document generation and pinpointing areas for potential improvement. This work marks a step toward a more effective framework for automating legal case documents drafting, paving the way for the reliable application of AI in the legal field. The dataset and code are publicly available at `https://anonymous.4open.science/r/CaseGen-3C46`.

## 1 Introduction

Legal case documents are the official records of judicial proceedings, containing factual determinations, legal rationale, judgment outcomes, and other relevant details (17; 14). The quality of legal case documents directly affects both judicial fairness and trial efficiency (23). Generally, drafting a high-quality legal case document involves extracting relevant information from extensive evidence, identifying key points of contention, and ensuring logical consistency. Legal professionals must devote significant time and effort, often spending dozens of hours to complete a legal case document (4; 28). With the explosive growth in legal cases, manually drafting legal case documents now faces pressure from tight deadlines and heavy workloads, making it challenging to balance efficiency and accuracy.

The rise of large language models (LLMs) presents a promising alternative to manually drafting legal case documents (13). These models, trained on vast text corpora, have shown a remarkable ability to understand and generate human-like text (2). Despite their potential, applying LLMs to generate legal case documents continues to present significant challenges. Legal case documents require a high level of professionalism and accuracy. However, probability-based LLMs are prone to hallucinations (26), which means they cannot guarantee the correctness and interpretability of their outputs. If an LLM generates low-quality or misleading legal case documents, it not only increases the workload of legal professionals but also may significantly undermine the fairness of the judgment (20).

The potential and risks of LLMs in generating legal case documents highlight the urgent need to evaluate their professional performance. However, there is currently no representative benchmark that covers all aspects of legal case documents generation. Existing datasets in general domains primarily focus on general text processing tasks, such as summarization and open-ended question

answering, providing limited guidance for specialized fields like law (11; 31). Furthermore, existing legal datasets concentrate on relatively straightforward tasks, such as judgment prediction (34) or legal case retrieval (21). These tasks are typically discriminative with limited output spaces, failing to fully capture the complexity and diversity involved in drafting real-world legal case documents.

To fill this gap, we propose CaseGen, a comprehensive benchmark for multi-stage legal case documents generation in the Chinese legal domain. Built on high-quality, real-world legal case documents and expert annotations, CaseGen includes 500 instances, each consisting of seven sections: Prosecution, Defense, Evidence, Events, Facts, Reasoning, and Judgment. It supports four key tasks: drafting defense statements, writing trial facts, composing legal reasoning, and generating judgment results. CaseGen provides a comprehensive evaluation platform for assessing the strengths and limitations of LLMs in generating legal case documents.

Specifically, CaseGen is unique from the following three perspectives:

1. **First Comprehensive Legal Case Documents Generation Benchmark.** To the best of our knowledge, CaseGen is the first Chinese benchmark designed to evaluate LLMs in legal case document generation. It covers all key stages, from the initial complaint to evidence and judgment, providing a complete framework for assessing LLM performance.

2. **Multi-Stage Generation Task Support.** Instead of directly generating entire case documents, CaseGen follows the structure and writing process of real-world legal case documents, designing multi-stage generation tasks. It includes four key tasks: drafting defense statements, writing trial facts, composing legal reasoning, and generating judgment results. Each task has its own writing logic and evaluation criteria, enabling a more detailed and nuanced assessment.

3. **Automated Evaluation Framework.** Relying on human evaluation for quality evaluation is both costly and time-consuming. To achieve efficient automated evaluation, CaseGen adopts the LLM-as-a-judge scoring approach (22). The LLM judges assign pointwise scores based on task-oriented criteria, referencing the ground truth and employing Chain-of-Thought (CoT) reasoning (33). Human evaluations confirm the effectiveness of this method.

We conduct a systematic evaluation of various commercial and open-source LLMs. The results show that current LLMs do not achieve satisfactory performance in legal case document generation. Additionally, human annotations show that our evaluation framework aligns closely with legal expert annotations. We also provide an in-depth analysis of the strengths and limitations of LLMs, highlighting key areas for future development.

## 2 RELATED WORK

### 2.1 LLMs FOR LEGAL APPLICATIONS

Recently, LLMs have profoundly impacted the legal domain, significantly enhancing the efficiency, accuracy, and scalability of legal services (13; 15). For instance, Daniel Martin et al. (12) demonstrate that GPT-4 successfully passes the Uniform Bar Examination (UBE), outperforming both previous models and human test-takers. This highlights its potential to enhance legal services and advance NLP in the legal domain. Despite the impressive performance, general LLMs encounter significant challenges in complex legal reasoning and specialized tasks, primarily due to their limited domain-specific legal knowledge. Therefore, researchers worked to improve the legal adaptability of LLMs through continue pretraining or fine-tuning. For instance, LexiLaw (16) enhances its expertise and performance in legal consultation and support through fine-tuning on legal domain datasets. ChatLaw (6) integrates knowledge graphs and manual curation to build a high-quality legal dataset for training MoE models, boosting the reliability and accuracy of AI-driven legal services.

### 2.2 BENCHMARKS IN THE LEGAL DOMAIN

LLMs have shown great potential in the legal domain. However, their inherent limitations emphasize the urgent need for comprehensive evaluation. In response, researchers have developed various evaluation criteria and benchmarks. For instance, LegalBench (10) is a collaboratively developed

legal benchmark comprising 162 tasks, designed to assess legal reasoning in English LLMs. Similarly, LawBench (8) and LAiW (7) leverage existing datasets to evaluate the Chinese legal LLMs, fostering community advancement. LexEval (20) presents a taxonomy of legal cognitive abilities and organized 14,150 tasks to systematically evaluate LLMs in the legal domain. Moreover, Li et al. (19) introduced LegalAgentBench, which provides 37 tools for interacting with external knowledge and evaluates LLM agents in the legal domain. Despite these advancements, a dedicated benchmark for legal case document generation is still lacking. This paper introduces CaseGen, which fills this gap by providing a comprehensive benchmark for multi-stage legal case documents generation in the Chinese legal domain.

## 3 PRELIMINARIES

In this section, we introduce the structure of legal case documents, which guides the design of our task. Specifically, unlike documents in general domain, legal case documents typically have a more structured format. Following the definition by Li et al., (14), legal case documents generally consist of five parts: **Procedure**, **Trail Fact**, **Reasoning**, **Judgment** and **Tail**. The Procedure section includes claims, defense statements, and the evidence lists submitted by both parties. The Trial Fact section presents the verified events as determined by the court. The Reasoning section explains how the court analyzes disputed issues, selects relevant legal rules, and applies them to the case facts. The Judgment section includes the court's final ruling and relevant legal provisions. The Tail section contains details such as the court's name, judge information, and other procedural formalities.

Each section of a legal case document follows distinct writing logic and evaluation criteria. For example, the Trial Fact section prioritizes a complete evidence chain and an accurate timeline, while the Reasoning section focuses on identifying key issues and applying legal rules correctly. These differences impose distinct demands on the LLM's understanding and reasoning abilities. Generating a complete case document in one step fails to properly evaluate the LLM's performance in generating each structural component. Therefore, we design the multi-stage generation task that aligns with the writing logic of legal case documents. This approach not only enables more precise evaluation of LLMs, but also provides a more reliable solution for practical legal AI applications.

## 4 CASEGEN

Developed from high-quality, real-world legal cases, CaseGen comprises 500 instances, each structured into seven distinct sections: **Prosecution**, **Defense**, **Evidence**, **Events**, **Trial Fact**, **Reasoning**, and **Judgment**. The Prosecution is a formal document filed by the plaintiff to initiate litigation, detailing the claims and supporting facts. The Defense is the responds to the Prosecution, in which the defendant challenges the plaintiff's claims and presents their own arguments. The Evidence includes all expert-annotated case-related evidence details, with each piece corresponding to an event in the trial facts. The Facts, Reasoning, and Judgment sections form the core components of a legal case document. A more detailed description can be found in Section 3. We provide a task sample of CaseGen in Figure 1.

### 4.1 TASK DEFINITION

CaseGen includes four key tasks: (1) drafting defense statements, (2) writing trial facts, (3) composing legal reasoning, and (4) generating judgment results. These tasks reflect different stages in the document creation process, each with its own writing logic and evaluation criteria, enabling a more precise and comprehensive assessment of LLMs.

### 4.1.1 DRAFTING DEFENSE STATEMENTS

The task of drafting defense statements involves systematically responding to the claims in the prosecution based on the provided evidence list. An effective defense should be clear and logically organized. Furthermore, it should directly address each claim while integrating relevant legal knowledge and supporting evidence.

- **Id**: 0
- **Title**: First-Instance Civil Judgment on Lease Contract Dispute between Li and Feng
- **Full_text:** … After accepting the case, this Court conducted a public hearing in accordance with the ordinary procedure…
- **Prosecution:** Claims: 1. that the defendant's conduct constitutes a fundamental breach of contractConfirm…
- **Defense:** The plaintiff's allegations are largely untrue. First, the plaintiff's claim that the respondent used another company's business license to induce other companies into signing lease contracts on a large scale is not true.
- **Fact:** Upon examination, it was ascertained that on July 11, 2010, the plaintiff and the defendant entered into a Factory Lease Contract…
- **Reasoning:** This Court finds that the factory leased by the defendant to the plaintiff lacks a property ownership certificate and the necessary construction approval formalities…
- **Judgement:** In conclusion, pursuant to [relevant legal provisions], the judgment is as follows…
- **Evidence:** {"Factory Lease Contract": "Party A: Li Date: July 11, 2010…", …}
- **Event:** {"Factory Lease Contract": "On July 11, 2010, Li and Feng entered into a Factory Lease Contract.",…}

Figure 1: An task example in CaseGen.

### 4.1.2 WRITING TRIAL FACTS

The task of writing trial facts can be defined as verifying the true course of events and identifying the key facts based on the provided evidence list, prosecution, and defense statement. Since the factual statements in the prosecution and defense may be incomplete or even contradictory, the court must evaluate the evidence to establish the trial facts. High-quality trial facts should be presented in a clear narrative structure, with a complete timeline and evidentiary chain. Furthermore, all information should be directly relevant to the legal proceedings, with unnecessary details kept to a minimum.

### 4.1.3 COMPOSING LEGAL REASONING

Legal reasoning refers to the process by which judges analyze case facts and apply legal principles to justify their rulings. High-quality legal reasoning should clearly identify all key issues in dispute and present the corresponding judicial perspectives. Since legal reasoning requires balancing multiple legal arguments and precisely applying legal provisions, it is one of the most challenging task of legal case documents generation.

### 4.1.4 GENERATING JUDGMENT RESULTS

Generating a judgment results involves formulating the final ruling based on established trial facts and legal reasoning. This section typically cites relevant legal articles and specifies the corresponding penalties. A well-crafted judgment must be legally sound, enforceable, and logically reasoned, ensuring judicial integrity and fairness.

Figure 2 illustrates the relationship between different tasks in CaseGen. To effectively prevent error accumulation, each subtask uses authentic documents as input rather than model-generated content. For example, the input for writing trial facts is the authentic defense statement, not the model-generated defense from the previous task. This multi-stage generation approach allows for

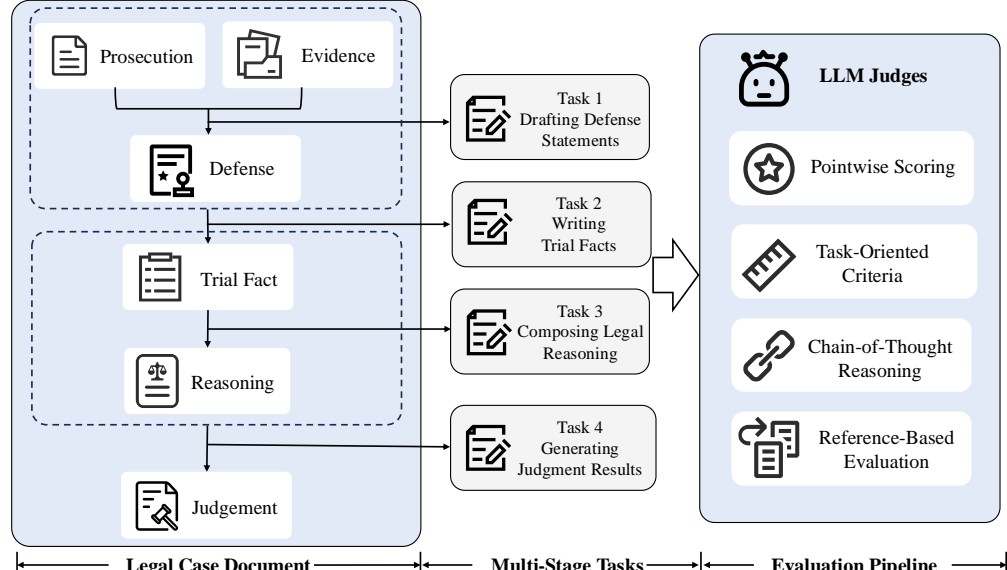

Figure 2: The overview of CaseGen. CaseGen includes four key generation tasks and uses LLMs-as-a-judge as the primary evaluation method.

a more precise evaluation of the strengths and weaknesses of the current LLM in legal document drafting tasks. Due to space constraints, additional task examples and the prompts used are provided in Appendix B.

## 4.2 DATA CONSTRUCTION

### 4.2.1 DATA SOURCE AND PROCESSING

CaseGen is built on high-quality legal documents. We collected hundreds of thousands of legal case documents from China Judgments Online [1] and implemented rigorous data filtering and processing techniques to ensure data integrity and quality.

Specifically, we first filter out cases where the fact section contains fewer than 50 chinese characters or involves simplified procedures, as these cases are considered too simplistic. Additionally, we exclude cases with incomplete structures or formatting errors to maintain data consistency. During filtering, we found that not all legal documents fully record both the plaintiff's claims and the defendant's defenses. Therefore, we carefully selected 50,000 cases that explicitly include both. To further enhance the diversity and representativeness of the dataset, we then use BGE-base-zh (35) to generate case embeddings and apply K-Means (3) clustering to group similar documents. From these clusters, we select 500 representative cases evenly as the core dataset for CaseGen.

Then, we utilized regular expressions and text parsing techniques to extract key structural information from legal case documents. The extracted data is structured in JSON format. For cases that are difficult to parse automatically, we manually extract the various sections and conduct thorough verification.

### 4.2.2 DATA ANNOTATION

Although high-quality legal case documents generally contain well-structured information, the full details of evidence are often not publicly disclosed. These case documents usually list the names of the submitted evidence without providing their content. To ensure data completeness and usability, we recruit legal experts to annotate the content of the evidence.

The annotation follows three core principles: (1) **Authenticity.** Since LLMs cannot independently verify the authenticity of evidence, all annotated evidence is authentic, excluding any uncertain or potentially falsified information. (2) **Completeness.** The annotated evidence must accurately align

---

[1]https://wenshu.court.gov.cn/

| Statistic | #Number |
|:---:|:---:|
| Total Legal Case Document | 500 |
| Avg. Full Case Length | 5,223 |
| Avg. Complaint Length | 1,187 |
| Avg. Defense Length | 1,100 |
| Avg. Fact Length | 1,057 |
| Avg. Reasoning Length | 1,241 |
| Avg. Judgement Length | 450 |
| Avg. Evidence per Query | 7.92 |
| Avg. Evidence Length | 706 |

Table 1: Basic statistic of CaseGen.

with the content of the legal case document, ensuring that the entire trail fact can be reconstructed from the provided evidence. (3) **Textual Representation.** All evidence is presented in textual form. For non-text evidence, such as audio recordings or images, experts provide descriptive textual summaries to ensure clarity and usability. Additionally, annotation experts need to convert litigation claims and defense arguments from the Procedure section into structured prosecution and defense statements. For more detailed annotation guidelines, please refer to Appendix E.

Our annotation team comprises five legal experts, all of whom have passed the National Unified Legal Professional Qualification Examination and possess a strong legal background. The team includes two male and three female experts, all based in China. To protect the rights and interests of annotators, we established legally binding agreements with all team members before the annotation work began. These agreements ensure compliance with legal standards and protect the experts' rights throughout the annotation process.

To ensure data quality, all annotators must complete comprehensive training. We first provide a detailed explanation of the task objectives, data formatting requirements, and applicable legal standards. Subsequently, some example cases are provided to help annotators understand the required format and standards. We provided several hours of in-depth training to ensure annotators fully understood the annotation standards. Following this, each annotator was required to complete five pilot annotation tasks. Our gold annotator, who hold a Ph.D. in law, conducted cross-check evaluations to review and verify the accuracy of the pilot annotations. Only annotators with an approval rate of 90% or higher were permitted to proceed with formal annotations.

For each annotated dataset, we employ a dual verification process using both LLMs and human experts. We first employ an LLM for automated review to verify evidence completeness. Then, legal experts conduct cross-checks to ensure legal compliance and accuracy. The detailed review process can be found in Appendix E. For each successfully reviewed example, we paid $10.95 to the legal annotators. A total of 500 examples were annotated, amounting to a total payment of $5,475.

## 4.3 DATA STATISTICS

After careful manual verification, CaseGen consists of four types of tasks, with each task containing 500 test samples. Table 1 presents the basic statistical information. Compared to general-domain texts, legal case documents are significantly longer. On average, each case contains 7.92 pieces of evidence, with each piece averaging 706 characters in length. Additionally, the generated texts can reach lengths of up to 1,000 characters. This poses a significant challenge for LLMs in handling long-text processing effectively.

## 4.4 EVALUATION PIPELINE

Evaluating legal case documents is a challenging task. Traditional evaluation metrics, such as BLEU (25), ROUGE (24), and BERTScore (38), fail to capture key aspects like fluency, logical coherence, and factuality. While human evaluation is reliable, it is time-consuming and labor-intensive, making it difficult to scale for large-scale assessments. Therefore, we adopt LLM-as-a-judge as the core evaluation method in CaseGen. Recently, LLMs have gained widespread recognition for their effectiveness as evaluators, achieving a high level of consistency with human annotations (22). Compared to traditional automated evaluation metrics, LLM-as-a-judge enables a more fine-grained, multi-dimensional assessment (5; 18).

| Model | Defense | | | Fact | | | Reasoning | | | Judgement | | |
|---|---|---|---|---|---|---|---|---|---|---|---|---|
| | ROU. | BS. | LLM | ROU. | BS. | LLM | ROU. | BS. | LLM | ROU. | BS. | LLM |
| LexiLaw | 6.18 | 62.38 | 1.17 | 8.16 | 59.70 | 1.18 | 8.27 | 67.65 | 2.36 | 13.20 | 66.17 | 2.22 |
| ChatLaw | 6.44 | 64.03 | 2.09 | 25.62 | 70.19 | 2.43 | 9.53 | 69.35 | 3.41 | 21.80 | 67.54 | 2.27 |
| GLM-4-flash | **25.28** | 74.07 | 4.26 | 39.59 | 75.05 | 3.82 | 19.71 | 71.57 | 5.01 | 26.28 | 72.71 | 3.42 |
| GLM-4 | 23.05 | 73.31 | 4.47 | 39.55 | 74.82 | 4.32 | 18.68 | 71.58 | 5.39 | **26.69** | **75.85** | 3.59 |
| Qwen2.5-72b-instruct | 22.39 | **75.45** | **4.97** | 46.32 | 76.47 | 4.58 | 23.50 | 71.33 | **6.19** | 19.45 | 74.23 | **4.46** |
| Llama-3.3-70b-instruct | 21.11 | 70.68 | 4.07 | 37.43 | 74.57 | 3.58 | 23.54 | 72.72 | 4.87 | 21.65 | 70.69 | 4.05 |
| GPT-3.5-turbo | 19.89 | 71.67 | 4.90 | 38.22 | 73.98 | 4.31 | 22.59 | 71.18 | 5.90 | 17.78 | 70.71 | 3.99 |
| GPT-4o-mini | 20.84 | 71.35 | 4.83 | 36.00 | 73.69 | 3.99 | 22.46 | 71.50 | 5.66 | 18.52 | 71.03 | 3.88 |
| Claude-sonnet | 23.60 | 73.31 | 4.91 | **53.03** | **77.92** | **4.75** | **25.16** | **72.74** | 5.77 | 25.62 | 77.00 | 4.00 |

Table 2: The main results of the four tasks in CaseGen. "ROU." represents the ROUGE-L score (%), "BS." stands for BERTScore (%), and "LLM" refers to the scores assigned by the LLM Judge. The best results are highlighted in bold.

However, evaluating legal case documents poses even greater challenges for LLM judges, requiring not only domain-specific expertise but also strict logical reasoning. Moreover, each section follows distinct evaluation criteria, further complicating the evaluation process. Following Wang et al., (30), we developed a multi-dimensional automated evaluation framework for legal case documents generation, ensuring both professionalism and reliability. As shown in Figure 2, the evaluation framework includes the following four key features:

**Pointwise Scoring.** We employ a pointwise scoring method, which offers greater flexibility compared to pairwise comparisons. Specifically, LLM judges perform a multi-dimensional analysis of the generated documents and assign a final score from 1 to 10, with higher scores indicating better quality.

**Task-Oriented Criteria.** Different sections of legal case documents require distinct evaluation criteria. To address these variations, we establish fine-grained evaluation criteria based on expert-defined standards, covering multiple dimensions such as accuracy, logical consistency, completeness, and legal applicability. For each task, we provide specific evaluation dimensions with detailed explanations to ensure LLM judges accurately reflects the quality of the generated documents. Additionally, we establish scoring standards for the LLM judges, with each 2-point increment representing a different rating level.

**Chain-of-Thought Reasoning.** To enhance the reliability of LLM judges, we incorporate Chain of Thought (CoT) reasoning, allowing the LLMs to assess the generated content step by step rather than assigning a score directly. Specifically, the LLM judge first compares the generated output with the reference answer, then assigns scores for each evaluation dimension, and finally consider all dimensions to determine the overall score.

**Reference-Based Evaluation.** Evaluating legal case documents requires extensive legal expertise. To address this, we adopt the reference-based evaluation approach, where the ground truth is provided as part of the input to the LLM judges. This allows the LLM to contextually compare the generated text with authoritative references, ensuring a more informed and precise evaluation.

More detailed explanations and examples are provided in Appendix C. We further validate the effectiveness of our evaluation framework through human annotations in Section 5.2.1. It is worth noting that we did not rely solely on LLM-as-a-judge as the only evaluation method. Instead, we combined it with traditional evaluation metrics to conduct a comprehensive automated assessment. In Appendix F, we conducted cross-validation experiments using different LLMs to verify the stability of our evaluation framework.

## 4.5 LEGAL AND ETHICAL CONSIDERATIONS

Due to the sensitivity of the legal domain, we have conducted a thorough review of this benchmark. All the open-source datasets we use are licensed. We have also carefully screened and filtered the datasets to avoid any content containing personal identifiable information, discriminatory material, explicit, violent, or offensive content. A more detailed discussion can be found in Appendix A.

## 5 EXPERIMENT

### 5.1 EXPERIMENTAL SETTINGS

We evaluated several popular commercial and open-source models, including GLM-4-flash (9), GLM-4 (9), Claude-3.5-sonnet, GPT-3.5-turbo (2), GPT-4o-mini (2), Qwen2.5-72B-Instruct (36), and LLaMA-3.3-70B-Instruct (29). Additionally, we assessed legal-specific LLMs, including ChatLaw (6) and LexiLaw (16).

To ensure reproducibility, we set the temperature of all LLMs to 0. All LLMs are evaluated with the same prompt to ensure a fair comparison. When the input text exceeds the LLM's maximum context window, we truncate the input sequence from the middle since the front and end of the input may contain crucial information. We use GPT-4o as the LLM judge to evaluate the performance of other LLMs. In addition to LLM Judge scores, we also provide ROUGE-L (24) and BERTScore (38) as reference metrics. Due to space limitations, more implementation details are provided in Appendix D.

### 5.2 MAIN RESULT

The performance comparison of different LLMs is presented in Table 2. We derive the following observations from the experiment results.

- **Legal-specific LLMs exhibit suboptimal performance.** Despite additional training on legal datasets, legal-specific LLMs such as Lexilaw and ChatLaw perform worse than general LLMs in legal case document generation tasks. This may be attributed to two key reasons. First, the performance of legal-specific LLMs may be limited by the constraints of their base models, which often lack the advanced comprehension and long-text processing capabilities of state-of-the-art general LLMs such as GPT-3.5 and Qwen2.5. For example, Lexilaw has a maximum input length of only 2048 tokens, which may lead to information loss when processing lengthy legal case documents, significantly impacting the quality of the generated context. Another possible reason is that continuous training on legal corpora may reduce the reasoning abilities inherited from the original base model, limiting its overall effectiveness in generating complex legal cases. This suggests that legal-specific LLMs need further optimization of training strategies to improve legal reasoning capabilities.

- **Open-Source LLMs Demonstrate Competitive Performance in Legal Case Documents Generation.** Compared to closed-source models like GPT-3.5-turbo and Claude-sonnet, open-source LLMs have achieved competitive performance in legal case documents generation tasks. Qwen2.5-72B-Instruct achieved the highest LLM judge scores in drafting defense statements, writing trial facts, and generating judgment results. These results highlight the potential of open-source LLMs as a viable alternative to commercial LLMs. With continued improvements, open-source LLMs are expected to play an increasingly important role in legal AI applications, making further exploration and development essential.

- **Existing LLMs Still Struggle with Legal Case Documents Generation.** Across multiple tasks evaluated by CaseGen, most LLMs achieve unsatisfactory scores (below 6 points), indicating that they fail to meet the basic quality standards required for legal case documents. This highlights the significant challenges that existing LLMs still face in handling complex legal reasoning. These LLMs often struggle to generate text that is not only legally precise but also logically coherent. Furthermore, it emphasizes the value and challenges of CaseGen as a benchmark for legal document generation, providing clear guidance for the future development of legal AI and specialized LLMs.

### 5.2.1 HUMAN EVALUATION ON CASEGEN

In this section, we recruit legal experts to evaluate LLM-generated texts and assess the consistency between LLM judges and human annotations. Due to cost limitations, we randomly select 50 cases from CaseGen. For each question, we obtain the response from three LLMs: Qwen2.5-72b-instruct, GPT-4o-mini, and Claude-sonnet, as these LLMs demonstrate competitive performance on CaseGen. Each LLM completes four tasks from CaseGen, generating a total of 600 samples to be evaluated.

| Model | Defense | Fact | Reasoning | Judgement |
|---|---|---|---|---|
| Qwen2.5-72b | **4.76** | 4.98 | **5.56** | **6.02** |
| GPT-4o-mini | 4.16 | 5.5 | 5.32 | 5.78 |
| Claude-sonnet | 4.66 | **5.98** | 5.46 | 5.64 |

Table 3: Results of Human Annotation. The best results are highlighted in bold.

| Metrics | LLM Score | Rouge-L | BERTScore |
|---|---|---|---|
| Kendall | **0.667** | 0.333 | 0.166 |
| Pearson | **0.726** | 0.264 | 0.239 |
| Spearman | **0.750** | 0.375 | 0.250 |

Table 4: The consistency between different automated metrics and human annotations. The best results are highlighted in bold.

We recruit three legal experts, all of whom have passed the National Unified Legal Professional Qualification Examination, to carry out the annotation tasks. We convert each sample into a input-response-reference triple and present it to human annotators. To prevent potential bias, annotators were unaware of which LLM generated the response, and the responses were provided in random order. The annotation criteria provided to the experts align with those given to the LLM judges, ensuring a fair comparison. We use the Kappa statistic (32) to measure the consistency and quality of the human annotations. The Kappa values (32) for the three annotators across the four tasks are 0.428/0.488/0.539/0.494, respectively, indicating the reliability of our annotations. We paid $0.21 for each annotation example, totaling $385.20.

Table 3 presents the results of the human annotations. For legal experts, the legal case documents generated by LLMs are still unsatisfactory (below 6 points). Even the most advanced LLMs still cannot generate legal case documents that are truly suitable for practical use. Moreover, we observe that legal experts gave slightly higher average ratings for the tasks of writing trial facts and generating judgment results compared to the LLM judges. This may be because legal experts can better understand the context and nuances of legal provisions, allowing them to make more accurate judgments based on real-world cases. On the other hand, LLM judges face limitations in accuracy and logical rigor when dealing with complex legal relationships and dynamic statutes, as they can only compare responses to reference answers. Further improvements are still needed in the performance of LLM judges within the legal field.

Then, we calculated Spearman's rank correlation coefficient (37), Kendall rank correlation coefficient (1), and Pearson correlation coefficient (27) between the automated metrics and human evaluation results. Since the evaluation dimensions vary across tasks, we calculated the consistency for each task separately and then averaged the results. Table 4 presents the consistency result. We observe that the LLM judge score demonstrates the highest level of consistency with human annotations, with a Spearman's rank correlation coefficient reaching 75%. In contrast, Rough-L, which relies on lexical matching, demonstrated lower consistency. BERTScore, which compresses context into vectors to calculate similarity, results in the loss of important details and thus demonstrates the lowest consistency with human annotations. In conclusion, our evaluation pipeline shows high consistency with human assessments, making it a reliable alternative for large-scale evaluations.

# 6 CONCLUSION

In this paper, we present CaseGen, the first comprehensive benchmark designed to evaluate LLMs in legal case documents generation task. CaseGen fills a critical gap by providing a robust framework for evaluating LLMs in multi-stage legal document generation. By covering all key stages of legal document creation—from prosecution to judgment—it enables a more nuanced evaluation of LLM performance in tasks that capture the complexities of real-world legal work. Additionally, CaseGen supports four key tasks: drafting defense statements, writing trial facts, composing legal reasoning, and generating judgment results. It offers both researchers and practitioners a means to identify strengths and weaknesses in current LLMs, laying the foundation for future improvements in automated legal case documents generation. In the future, we will further refine the automated evaluation framework for legal documents generation to achieve more accurate and comprehensive assessment results.

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

# A DISCUSSION

In this section, we provide a detailed discussion on the limitations of CaseGen, its broader impact, licensing, and ethical considerations.

## A.1 LIMITATION

While CaseGen advances the evaluation of LLMs in the legal domain, several limitations still require further refinement. First, CaseGen is built on Chinese legal cases, which means it may not capture the diversity and complexity of legal systems worldwide. Different countries and regions have distinct legal frameworks, litigation procedures, and document formats. CaseGen is currently unable to evaluate document generation across all legal environments. Additionally, while our automated evaluation framework has been validated by legal experts, it cannot fully replace the nuanced professional judgment. The evaluation results generated by the LLM may be influenced by the LLM's inherent limitations, especially in complex legal reasoning tasks. Furthermore, LLM judges are vulnerable to adversarial attacks, potentially compromising the stability and reliability of the evaluations. In future work, we will expand the dataset to include legal case documents from more countries and regions, enhancing the applicability and comprehensiveness of the research. Additionally, we will further explore combining LLM judges with traditional metrics to improve the robustness of the evaluations.

## A.2 BROADER IMPACT

The research on CaseGen has significant implications for the digital transformation of the legal field. From a professional perspective, it enhances the efficiency of drafting legal documents. By reducing repetitive tasks, legal professionals can focus more on high-value analytical work, improving both the quality and efficiency of legal services. Additionally, the multi-stage generation framework advances AI application in law. From a technical perspective, it provides concrete methods and pathways for evaluating legal AI systems, filling the gap in current evaluation benchmarks within legal AI. It not only helps legal professionals better understand and assess the quality of LLM-generated documents but also lays the groundwork for evaluating similar technologies in the future.

To ensure fairness and transparency, CaseGen undergo strict ethical review and broad oversight. However, we urge caution against over-relying on AI-generated content. The goal of CaseGen is to enhance the efficiency of legal professionals, not to replace human legal expertise. The unique complexity of legal judgment still requires human insight, with AI serving as a supportive tool rather than a substitute for professional analysis. The technologies and evaluations related to CaseGen are for reference, and real-world legal applications still require human judgment.

In summary, CaseGen strongly supports the digital transformation of the legal domain by driving the automation of legal case document generation and enhancing judicial transparency and efficiency. However, we emphasize that AI must be applied responsibly and carefully to ensure it upholds fairness and reliability in the judicial process.

## A.3 LICENSE

The case documents and legal articles in CaseGen come from publicly accessible legal resources that comply with relevant legal and ethical standards. These resources are provided in compliance with applicable open access legal information guidelines, ensuring that their inclusion in the benchmark raises no legal or ethical concerns. Although the copyright of these materials remains with the respective government institutions, they have been publicly released and authorized for public use. Users must comply with all relevant laws and regulations when using this data. Furthermore, CaseGen fully complies with privacy protection and information security standards during data processing, especially when handling sensitive case information. All case documents have personal identifiers removed to ensure the privacy of the parties involved is protected.

CaseGen is released under the CC BY-NC-SA 4.0 license, allowing for noncommercial academic use with proper attribution. Commercial use requires additional authorization from the Document and Archives Department of the Supreme People's Court. Users must comply with China's Personal Information Protection Law and Cybersecurity Law while using the data in a reasonable and compliant

manner. The CaseGen team assumes no responsibility for any violations of these laws and regulations. We are dedicated to ensuring the legality and ethical standards of CaseGen and are happy to address any copyright concerns. If you believe that CaseGen contains any content that violates your copyright, please contact us, and we will promptly take action to resolve the issue and remove the content.

### A.4 ETHICAL CONSIDERATIONS

The development and release of CaseGen have always followed strict ethical standards, ensuring compliance with relevant laws, regulations, and guidelines. During preprocessing, we implemented strong measures to safeguard personal privacy, anonymizing any data that could identify individuals. All data has been thoroughly reviewed to ensure compliance with privacy protection laws, data security regulations, and ethical research guidelines. Legal experts have also reviewed the dataset to ensure it is free from harmful, offensive, or discriminatory material. We are committed to preventing any content that could harm individuals or groups and have taken steps to filter out discriminatory, violent, explicit, or offensive content.

## B MORE TASK DETAILS

Tables 5 to 8 present the prompts used for different tasks. For each task, the input information includes all the relevant content required to complete the task, All input information is based on ground truth, not LLM-generated content. This ensures that LLMs can generate or reason based on accurate information. Additionally, we emphasize the writing requirements corresponding to each task in the prompts to ensure that LLMs generate context that are both accurate and logically consistent.

## C MORE EVALUATION DETAILS

### C.1 EVALUATION CRITERIA

In CaseGen, each task is evaluated using distinct criteria, as shown in Table . These criteria are carefully developed through discussions with legal experts to reflect the unique emphasis of each section within legal case documents.

### C.2 INSTRUCTION

In Table 10, we present a prompt template that uses LLMs-as judges for evaluation. This prompt includes the evaluation criteria, chain-of-thought reasoning step, scoring stardards, and output format requirements.

## D MORE IMPLEMENTATION DETAILS

In our experiments, the versions used for Claude-sonnet/GPT-3.5/GPT-4o-mini are claude-3-5-sonnet-20241022/gpt-3.5-turbo-1106/gpt-4o-mini-2024-07-18, respectively. Lexilaw uses the Lexi-Law_Finetune version, while Chatlaw uses the Chatlaw-13B version.

During the evaluation process, we also report both ROUGE-L and BERTScore. Since CaseGen is a Chinese legal dataset, BERTScore was initialized using chinese-bert-wwm [2]. All experiments presented in this paper are conducted on 8 NVIDIA Tesla A100 GPUs.

## E GUIDELINES FOR EXPERT-ANNOTATION

To ensure the quality, consistency, and reliability of the CaseGen, we implement a strict verification and annotation process based on the following principles and standards. We have hired legal experts to ensure that all annotations meet the highest standards of legal accuracy and relevance.

---

[2]https://huggingface.co/hfl/chinese-bert-wwm

Document Structure Extraction Check: Annotators first check the accuracy of the Fact, Reasoning, and Judgment sections structure extraction. If any errors are found, annotators must re-extract the relevant content from the full document.

Evidence Content Annotation: We provide annotators with a list of the evidence mentioned in the legal case document. Annotators must first verify the authenticity of the evidence. Then, they annotate the evidence content according to the facts presented in the Fact section. Special attention must be paid to ensure that the evidence content strictly matches the Fact section, such as contract signing dates, specific clauses, and other details. Once the evidence annotation is complete, annotators need to check if the evidence covers all the facts in the Fact section. In other words, they must verify whether all relevant facts can be deduced from the available evidence. If certain facts cannot be deduced from the existing evidence, annotators should supplement the missing evidence based on their professional knowledge. To ensure the quality of evidence annotation, annotators must also mark the corresponding Event for each piece of evidence, ensuring that all components of the fact are clearly outlined. Annotators must ensure that the relationship between Fact and Evidence is clear and accurate. Each piece of evidence must support a specific event, and this connection should be clearly presented in the legal case document. If any evidence fails to support the relevant events, annotators must note the reason and re-annotate the evidence.

Prosecution and Defense Annotation: Annotators must extract and revise the complaint and defense sections from the document, ensuring that these parts conform to the formal legal document format and language requirements.

Handling Uncertainties and Doubts: If annotators encounter uncertainties during the annotation process, they should take the following steps: (1) Consult Legal Experts: Annotators should consult authoritative legal documents, terminology glossaries, or directly seek advice from legal experts to resolve any unclear points. (2) Transparent Documentation: All decisions made during the annotation process must be clearly recorded, especially those made after consulting experts, to ensure transparency and consistency.

Feedback Mechanism: Annotators are encouraged to provide ongoing feedback, suggest improvements to the annotation process, or highlight challenges encountered during annotation. Based on annotator feedback, the annotation guidelines will be regularly reviewed and updated to meet the evolving needs of the CaseGen and improve annotation quality.

Review and Correction Mechanism: Once the annotations are complete, they undergo at least two rounds of independent review. (1) Evidence Integrity Check: In the first round of review, we use the Qwen-2.5-72B-Instruct to check the completeness of the evidence. Specifically, the LLM first verifies if the evidence list and content cover all the facts mentioned in the Fact section. Then, the LLM checks each piece of evidence to ensure it aligns with the event. These two checks are discriminative and well-suited for LLMs, ensuring high accuracy. (2) Legality and Multi-Angle Validation: The second round of review is conducted by legal experts, who perform multi-angle validation to ensure that the annotations align with actual legal standards and the specifics of the case. Experts will also review the legal validity of the annotations to ensure they comply with applicable laws and regulations. Any issues or disputes during the annotation process should be discussed in expert meetings to reach a unified standard and decision, ensuring both accuracy and legal compliance. Annotators must revise based on the review results and submit the updated annotations for a second round of review.

Final Data Quality Assurance: Before the final submission of the dataset, all annotations undergo a final check by a quality assurance team. This check includes verifying the completeness, accuracy, consistency, and legal relevance of the annotations. Only when the dataset fully meets the quality standards will it proceed to the next stage of processing.

## F    ROBUSTNESS OF THE EVALUATION FRAMEWORK

Using LLMs as evaluators may introduce bias, especially when the evaluation model shares architectural similarities with the models being evaluated. This overlap can lead to unintended favoritism or scoring deviations, undermining evaluation objectivity. In this section, we conduct a more in-depth empirical investigation into the robustness of our evaluation framework. Specifically, we design a 5-fold cross-validation experiment on the full benchmark dataset, focusing on the three models: GPT-3.5-turbo, GPT-4o-mini and Qwen2.5-72B-Instruct.

---

**Task: Drafting Defense Statements**

---

Prompt: You are a legal expert proficient in drafting defense statements. Based on the following prosecution and relevant evidence, please draft a detailed and rigorous defense statement. The defense statement should clearly and comprehensively address the plaintiff's allegations, supporting your position with legal arguments and evidence.

[Prosecution]
{Prosecution Content}

Please carefully read the prosecution and respond to each allegation with reference to the following relevant evidence. Ensure that each accusation is addressed individually in the defense and provide valid defense arguments.

[Evidence]
{Evidence Content}

When drafting the defense statement, please pay attention to the following points:
1. Strictly adhere to the formatting and regulatory standards for legal documents.
2. Ensure clear logic and well-structured arguments in the defense.
3. Use formal, accurate, and objective language, avoiding subjective assumptions or unnecessary emotional tones.
4. Accurately cite relevant legal provisions to ensure the legality and authority of the defense statement.

Defense Statement:

---

Table 5: The prompt template for drafting defense statements (translated from Chinese).

To ensure a diverse evaluation perspective, we introduce three independent evaluators with distinct model architectures: GPT-4o, Qwen3-235B-A22B, Deepseek-R1. These evaluators independently score two tasks: Defense Generation and Fact Verification. As shown in table 11, Qwen2.5-72B-Instruct consistently outperforms GPT-4o-mini across all evaluators and both tasks. Moreover, the relative ranking between models remains stable, regardless of which evaluator is used. This finding suggests that any potential evaluator-specific bias—such as that from GPT-4o—does not significantly affect the overall benchmarking results, even though GPT-4o is involved both as an evaluator and as a model being evaluated. We attribute this robustness to the design of our evaluation framework, particularly the integration of Chain-of-Thought (CoT) prompting and reference-based scoring. Together, these components contribute to enhancing the objectivity and stability of LLM-as-a-judge evaluation.

## G  USE OF LARGE LANGUAGE MODELS

Large language models were utilized in this work as general-purpose assistive tools. Specifically, LLMs were employed in two ways: (1) as part of our evaluation framework to provide comparative responses or baseline outputs for qualitative assessment, and (2) to assist in refining the linguistic clarity and expression of the manuscript during the writing process. At no point were LLMs used for ideation, experimental design, or generation of scientific content. All results, claims, and analyses presented in the paper were conducted and validated by the authors, who take full responsibility for the content.

**Task: Writing Trial Facts**

Prompt: You are a legal expert skilled in writing trial facts. Based on the following prosecution, defense statement, and evidence materials, you need to synthesize the factual descriptions from all the evidence to generate the "Trial Facts" section of a legal document.

[Prosecution]
{Prosecution Content}

[Defense Statement]
{Defense Content}

[Evidence]
{Evidence Content}

Please generate the "Trial Facts" section of the legal document based on the above content. When writing trial facts, please pay attention to the following points:
1. Comprehensive content: It should include all important factual information, leaving out no key details.
2. Accurate expression: Ensure that all factual descriptions are objective, fair, and clear, avoiding subjective assumptions or inaccurate descriptions, and fully reflecting the true nature of the case.
3. Clear structure: The facts should be presented in a well-organized manner, either chronologically or logically.
4. Logical coherence: Ensure the logical relationships between the facts are clear and reasonable, avoiding contradictions or disjointed narratives.

Trial Facts:

Table 6: The prompt template for writing trial facts (translated from Chinese).

**Task: Composing Legal Reasoning**

Prompt: You are a legal expert skilled in composing legal reasoning. Based on the following prosecution, defense statement, and facts, you are now required to write the reasoning section of a legal judgment.

[Prosecution]
{Prosecution Content}

[Defense Statement]
{Defense Content}

[Trial Fact]
{Trial Fact Content}

Please conduct a legal analysis and reasoning from the perspective of the court to derive the reasoning basis for the judgment. Please note the following requirements:
1. Comprehensive analysis: Based on the established facts, complaint, and evidence, analyze each point of dispute in the case, identify the core issues, and ensure a complete presentation of the case.
2. Fact and law integration: For each established fact, combine relevant legal provisions or judicial interpretations for analysis, explaining the logical basis for the application of the law.
3. Logical coherence: Ensure that the reasoning process is clear, rigorous, and smooth, with no gaps or leaps between sections
4. Neutral and objective: Reason from the perspective of the court, avoiding subjective or emotional language, and ensuring that the analysis is objective and fair.

Legal Reasoning:

Table 7: The prompt template for composing legal reasoning (translated from Chinese).

**Task: Generating Judgment Results**

Prompt: You are a legal expert skilled in generating judgment results. Based on the following facts and legal reasoning, you are now required to write the judgment result for the case. The judgment result should include the legal provisions cited and the outcome.

[Trial Facts]
{Trial Facts Content}

[Legal Reasoning]
{Legal Reasoning Content}

Please write a detailed judgment result, ensuring that relevant legal provisions are cited to support the judgment conclusion, and the reasoning is rigorous and the language is formal. The judgment result should clearly reflect the legal provisions used and explicitly state the final ruling of the case.

Judgment Result:

Table 8: The prompt template for generating judgment result (translated from Chinese).

| Criteria | Description |
| --- | --- |
| **Task: Drafting Defense Statements** | |
| Factuality | The legal facts in the defense statement should be accurate, and the relevant facts should be consistent with the reference defense statement, supported by sufficient evidence. |
| Legal accuracy | The characterization of legal relationships should be precise, with correct references to relevant laws, regulations, and judicial interpretations. |
| Logical Coherence | The structure of the defense statement should be clear, with a reasonable and coherent reasoning process that aligns with legal argumentation logic and debate logic. |
| Completeness | The defense statement should provide a comprehensive and appropriate response to the plaintiff's claims, covering all key points. |
| **Task: Writing Trial Facts** | |
| Factuality | The trial facts should be accurate and consistent with the reference answer, and must not contain any errors, omissions, or fabricated content. |
| Relevance | The trial facts should present a clear and complete chain of evidence, with a well-defined relationship between the evidence and the facts, ensuring that all information is directly relevant to the case proceedings. |
| Logical Coherence | The trial facts should have a clear structure and reasonable reasoning, ensuring that the narrative is logically coherent and follows the logical sequence of establishing legal facts |
| Completeness | The trial facts provides sufficient information and details, with no important facts omitted. |
| **Task: Composing Legal Reasoning** | |
| Dispute Accuracy | The legal reasoning section should accurately outline the core issues in dispute, aligning with the reference answer, and must not omit or incorrectly summarize the points of dispute. |
| Legal accuracy | The characterization of legal relationships should be precise, with correct references to relevant laws, regulations, and judicial interpretations. |
| Logical Coherence | The legal reasoning should be clearly structured and logically rigorous, adhering to the norms of legal reasoning, ensuring coherence and reasonableness in the argumentation. |
| Completeness | The legal reasoning should provides sufficient information and details, with no important points of dispute omitted. |
| Ethicality | The legal reasoning should comply with the requirements of legality and rationality, and must not contain discriminatory, prejudicial, or harmful content. |
| **Task: Generating Judgment Results** | |
| Judgment Accuracy | The judgment result is correct, aligning with the facts of the case and relevant legal provisions, thereby ensuring the legality and rationale of the judgment conclusion. |
| Legal Accuracy | The legal provisions cited in the judgment result are accurate and complete, aligning with the case context and legal framework, ensuring the rigor of the legal basis. |

Table 9: Evaluation criteria for different tasks (translated from Chinese).

---

Instruction Template used in LLM-as-a-judge.

You are an experienced legal expert specializing in assessing the quality of legal case documents. Please objectively evaluate the defense statement written by the AI assistant in the role of a fair and rigorous judge. When evaluating, you should score based on the following four key dimensions: {Task-Oriented Criteria}

We will provide the following materials: The prosecution, a high-quality reference defense statement, and the defense statement written by the AI assistant.When starting your evaluation, youneed to follow the reasoning steps below:
1. Compare the AI assistant's defense statement with the reference answer, pointing out the short-comings of the AI assistant's answer and explaining them in detail.
2. Evaluate the AI assistant's defense statement according to the dimensions mentioned above, giving a score from 1 to 10 for each dimension.
3. Based on the scores for each dimension, calculate the overall score for the AI defense statement (1-10 points).
4. Your scoring should be as strict as possible, and you must follow the scoring rules below: The higher the quality of the response, the higher the score.

Scoring Stardards:
If the defense statement includes irrelevant content, contains obvious factual or legal relationship errors, or generates harmful content with a large amount of unverified or false facts, the overall score should be 1-2 points.
If the defense statement does not contain serious errors but has issues with the characterization of legal relationships or fails to adequately address key points in the complaint, failing to meet basic defense requirements, the overall score should be 3-4 points.
If the defense statement generally meets defense requirements, with accurate facts and legal relation-ships but is average in terms of logical consistency and completeness, the overall score should be 5-6 points.
If the defense statement is close to the reference answer in quality, performing well in each evaluation dimension with no obvious flaws, the overall score should be 7-8 points.
If the defense statement is significantly better than the reference answer, fully responding to the claims, and performing almost perfectly across all evaluation dimensions, it should receive a score of 9-10 points.

As an example, the reference answer could receive an overall score of 8 points.
Please provide detailed evaluation comments during scoring. After each dimension score, make sure to provide an explanation. All scores should be integers. Finally, return the evaluation results in the following dictionary format:
{{"Factuality": score, "Legal Accuracy": score, "Logical Coherence": score, "Completeness": score, "Overall Score": total score}}

[Start of Prosecution]
{Prosecution Content}
[End of Prosecution]

[Start of Reference Defense Statement]
{Reference Defense Statement Content}
[End of Reference Defense Statement]

[Start of AI Assistant's Defense Statement]
{AI Assistant's Defense Statement Content}
[End of AI Assistant's Defense Statement]

Please begin the evaluation:

---

Table 10: The Instruction Template for LLM-as-a-Judge (translated from Chinese).

Table 11: Cross-evaluation results on Defense and Fact tasks using three different evaluators

| Model | Defense | | | Fact | | |
|---|---|---|---|---|---|---|
| Evaluator | Qwen3-235B | Deepseek-R1 | GPT-4o | Qwen3-235B | Deepseek-R1 | GPT-4o |
| Qwen2.5-72B-Instruct | 4.17 | 5.16 | 4.97 | 3.49 | 4.68 | 4.58 |
| GPT-3.5-Turbo | 3.98 | 5.03 | 4.83 | 3.14 | 4.29 | 4.31 |
| GPT-4o-mini | 3.89 | 4.93 | 4.91 | 3.03 | 4.27 | 3.99 |

