# OpenReview forum: "CaseGen: A Benchmark for Multi-Stage Legal Case Documents Generation"
_ICLR.cc/2026/Conference — Submitted to ICLR 2026_

### Official Review · Reviewer_WtDq · 2025-10-30

**Soundness:** 2
**Presentation:** 2
**Contribution:** 2
**Rating:** 2
**Confidence:** 4

**Summary:**

This paper introduces "CaseGen," a new benchmark for legal case document generation, which splits the complex process into four subtasks: Drafting Defense Statements, Writing Trial Facts, Composing Legal Reasoning, and Generating Judgment Results. The authors contribute a dataset of 500 cases annotated by legal experts and propose an LLM-as-judge evaluation method as an alternative to metrics like BLEU and ROUGE.

**Strengths:**

1. The paper addresses an important and practical challenge in the legal domain.
2. The creation of an expert-annotated dataset is a valuable contribution.

**Weaknesses:**

My major concerns are about clarity and evaluation. The descriptions of the dataset construction and task definitions are ambiguous, making the work difficult to understand. More critically, the reliability of the evaluation is questionable; the reported human inter-annotator agreement (Kappa < 0.5) is too low to establish a reliable "ground truth", which in turn undermines the validity of the proposed LLM-as-judge. Detailed Comments are shown below:

1. Ambiguity in Case Selection: The procedure for selecting the 500 representative cases is underspecified. The authors state they used K-means clustering but omit crucial details: How many clusters were generated? What was the distribution of cases across clusters (e.g., were sizes balanced)? What sampling strategy (e.g., random) was used to select cases from these groups? The paper should also justify why clustering was used over a more interpretable method for ensuring diversity, such as sampling cases based on their legal charges.

2. Missing Annotation Consistency: The paper does not report the inter-annotator agreement (IAA) scores from the initial dataset annotation phase (Section 4.2.2). This metric is essential for assessing the quality and reliability of the ground-truth data itself.

3. Lack of Data Filtering: The decision to keep all 500 initially selected cases is questionable. The authors note that some cases involved uncertain evidence or non-textual evidence (audio, images). Such cases are poor candidates for a text-generation benchmark and should have been filtered out to ensure data quality and task focus.

4. Ambiguous Task Definitions: The definitions of the four subtasks are unclear, particularly regarding their specific inputs. For example, in Task 3 (Composing Legal Reasoning), it is not specified whether the input includes only the trial facts or also the original prosecution, evidence, and defense statements. Besides, Figure 2 appears to be misleading. It depicts the input for Task 2 as solely the "Defense," whereas the text in Section 4.1.2 states the input is the "evidence list, prosecution, and defense statement." This inconsistency should be resolved.

5. Unsupported Motivation for Task Decomposition: The paper claims that splitting the generation process into four subtasks is necessary because the end-to-end task is too complex for current LLMs. This central claim is not substantiated with any empirical evidence. To justify this design choice, the authors should provide results from a baseline experiment showing a state-of-the-art LLM failing at the complete, end-to-end generation task.

6. Incomplete Dataset Statistics: Table 1, which presents the statistics for the CaseGen dataset, only provides average lengths. To give readers a comprehensive understanding of the dataset, this table should be expanded to include other key descriptive statistics, such as the minimum, maximum, and standard deviation for document lengths.

7. Low Human Agreement in Evaluation: The paper reports a human annotator's Kappa score of "almost below 0.5" for the evaluation task. A Kappa in this range (often considered "slight" or "fair" agreement) is insufficient to establish a reliable ground truth. If human experts cannot consistently agree on the quality of the generated text, it is difficult to trust the validity of an LLM-as-judge method that is calibrated against this inconsistent standard.

**Questions:**

Please see the weaknesses.

---

### Official Review · Reviewer_ZKfR · 2025-11-01

**Soundness:** 2
**Presentation:** 3
**Contribution:** 3
**Rating:** 4
**Confidence:** 3

**Summary:**

This paper aims at benchmarking legal case document generation across an entire lifecycle of the legal process: from drafting defense statements, writing trial facts, composing legal reasoning, to generating judgment results. The paper also proposes an evaluation pipeline using LLM as a judge, backed up by human annotations.

**Strengths:**

1. I really appreciate the great effort in setting up and collecting human annotations. I love your attempts to add rigor to your evaluations and how you also filter out worse annotations to make sure your data is more reliable.
2. I also really love your problem formulation, because it studies a life cycle of legal document generation, which is something novel and the existing 'comprehensive' benchmarks might be missing - I think they study different stages of legal case document generation separately and then aggregate analysis - but not with a focus on an entire lifecycle. So I think this work adds a nice complement and an original angle to the legal NLP research community.

**Weaknesses:**

1. There are some presentation issues that I think are detrimental to the quality of the paper. First and foremost it's the citation format that needs to be fixed. I don't think ICLR cites by numbers? I think the convention is to go with \citet or \cite in latex. Second, Figure 1 offers a decent qualitative example, but I fail at connecting the illustration to figure 2. Also, you have four tasks for your set-up, and I am finding it hard to understand what those four tasks are based on Figure 1 alone. I think there should be significant modifications.

2. I appreciate you releasing your inter-annotator agreement level - it is low (~50%) and I really think that however reduces the rigor of your eval - because it shows that the tasks are not that verifiable, even by 'gold' standards. So I hope to see you can analyze why the agreement level is low - just a few in-depth examples would give us much information and figure out the verifiability of this task.

3. As I said, I think the problem in this paper is original and interesting. However, I think what's missing is that you can also study how different components relate to each this longitudinal process, like tracking the long context of the language model and see it can improve the overall generation. Maybe some ablation study can help but it feels more like a future work to me.

**Questions:**

I would really appreciate if you can clarify the low inter-annotator agreement level and provide a few failure mode analysis, making some comments on how fundamentally verifiable this task is.

---

### Official Review · Reviewer_2oBZ · 2025-11-01

**Soundness:** 3
**Presentation:** 3
**Contribution:** 3
**Rating:** 4
**Confidence:** 4

**Summary:**

The paper introduces CaseGen, a benchmark for multi-stage legal case document generation in the Chinese legal domain, on 500 cases sampled and annotated by legal experts. The paper used LLM-as-a-judge evaluation framework further validated by human experts. The paper tests many generic and domain-specific LLMs on the proposed benchmark.

**Strengths:**

The dataset CaseGen, which the authors claim is the first Chinese benchmark designed to evaluate LLMs in legal case document generation. The number of annotated documents is 500, which is a reasonable number given that legal annotators are not easily available. The annotation spans five sections -- Prosecution, Defense, Evidence, Events, Trial Fact, Reasoning, and Judgment.

The authors extensively discuss evaluation dimensions and how each generated document must be evaluated on several grounds, depending on the structure of a legal document.

The dataset enables the evaluation of four tasks: (1) drafting defense statements, (2) writing trial facts, (3) composing legal reasoning, and (4) generating judgment results.

The human annotation process is discussed in detail.

**Weaknesses:**

The need for LLM-as-a-judge is not clear if qualified pool of human experts is already present.

Human inter-annotator agreement is not reported.

More ablations could have been tried, e.g., with temperature and other parameters.

Truncation of the input may have led to serious loss of context. Instead, the authors could have fed the documents in chunks. Length is a major challenge for legal documents, and this issue needed to be handled instead of being bypassed.

**Questions:**

“Legal-specific LLMs exhibit suboptimal performance.” – requires additional experiments to provide a detailed understanding of points of failure. It is understandable that a legal-specific LLM trained on a task-space that bears little to no resemblance to the task of case document generation (where the demands from LLMs are significantly higher) falters. An elaborate set of observations established through experiments can enhance the value of the paper and potentially help future research.

Direct reporting of observations on aspects such as how the size of a model impacts reasoning and other capabilities in the task (even better if it is done per section). Can a sub-10B model (LLaMA 3.1 8B or Qwen 2.5 7B) perform competitively?

How were annotation disagreements resolved?

How was the performance of the truncated documents when compared to the shorter ones?

---

### Official Review · Reviewer_uYkB · 2025-11-01

**Soundness:** 3
**Presentation:** 2
**Contribution:** 2
**Rating:** 2
**Confidence:** 4

**Summary:**

CaseGen presents a benchmark for producing Chinese legal case documents using LLMs. As the title of the paper suggests they produce these legal case documents in states, progressively giving the previous stage output in the prompt for the next. They propose using LLM as a judge to evaluate the documents produced by different LLMs. They then compare LLM-as-a-judge evaluation with human evaluation and conclude that LLM-as-a-judge is sufficient.

**Strengths:**

S1. While benchmarks for few legal tasks like judgement prediction, case similarity prediction etc exist, this paper attempts to produce a benchmark for long text generation. The long text generation being the specific case documents that are used in the Chinese court system.
S2. The authors have taken the trouble to engage legal experts to annotate the case documents produced by different models.
S3. The multi-stage case document generation and using prompt styles like chain of thought are reasonable.

**Weaknesses:**

W1. The case document discussed in the paper seems specific to the court system in China and it is not obvious how this dataset helps with structured long text generation using LLMs. The examples provided and the paper themselves do not explain why this is a complex task beyond talking about the accuracy requirements in this domain, hallucinations etc. Given the most recent models have become quite good at generating long text, the particular gap that this benchmark is filling is not very apparent.

W2. The multi-stage document generation seems a reasonable solution. However, how this is helping with correcting factual errors is not clear. One could assume getting precise facts of the case - say a number, date, amount of money - using a retrieval system is more fool proof. Such a solution could be implemented using agents and tools. There isn't much discussion on how this work gets the facts correct and not just the format of the document correct.

W3. The experimental observation that specialized legal models don't compare well with bigger models is understandable. But it is not clear why claude sonnet or GPT 4o-mini struggle against qwen-72b and llama-70b models, especially on the reasoning tasks. This is a bit counter intuitive. Claude's superior performance on facts might highlight the need for retrieval based facts mentioned in W2 above.

**Questions:**

1. Did you try implementing your casegen solution using agents and tools?
2. Did you consider any adversarial ideas, where you use the generated text to produce summary and then compare with a summary from human generated document. Just curious about this. It's fine if this is not a valid evaluation method in your opinion.

**Details Of Ethics Concerns:**

The authors themselves mention the ethical concerns in generating a dataset of this nature which could accidentally involve details of real world cases. There have been concerns about the use of models in legal systems and how these case documents affect the legal process in their court system is a concern. However I'm not sure if a ethics review is needed, since these are well known issues in legal benchmarks and the authors also talk about these concerns.

---

### Meta-Review · Area_Chair_PVbo · 2026-01-06

**Summary:**

In this submission, the authors propose a benchmark for multi-stage legal case document generation, which supports several critical tasks in this field. Although the reviewers acknowledged the significance and social meaning of this work, they have several concerns about the quality of annotation, the rationality of the evaluation metrics, the quality control of data collection, and the paper's writing quality. After reading the submission, I think the concerns are reasonable, and thus the paper requires one more round of polishing and review.

**Reviewer Concerns:**

The authors did not provide any feedback in the rebuttal phase, so none of the concerns were resolved.

**Reviewer Scores:**

The reviewers would have maintained their scores because of receiving no feedback.

---

### Decision · Program_Chairs · 2026-01-26

Reject